

# The French press: a repeatable and high-throughput approach to exercising zebrafish (*Danio rerio*)

Takuji Usui[1,2], Daniel W.A. Noble[1,2], Rose E. O'Dea[1,2], Melissa L. Fangmeier[1,2], Malgorzata Lagisz[1], Daniel Hesselson[2,3] and Shinichi Nakagawa[1,2]

[1] Evolution & Ecology Research Centre and School of Biological, Earth and Environmental Sciences, University of New South Wales, Sydney, Australia
[2] Diabetes and Metabolism Division, Garvan Institue of Medical Research, Sydney, New South Wales, Australia
[3] St. Vincent's Clinical School, University of New South Wales, Sydney, New South Wales, Australia

## ABSTRACT

Zebrafish are increasingly used as a vertebrate model organism for various traits including swimming performance, obesity and metabolism, necessitating high-throughput protocols to generate standardized phenotypic information. Here, we propose a novel and cost-effective method for exercising zebrafish, using a coffee plunger and magnetic stirrer. To demonstrate the use of this method, we conducted a pilot experiment to show that this simple system provides repeatable estimates of maximal swim performance (intra-class correlation [ICC] = 0.34–0.41) and observe that exercise training of zebrafish on this system significantly increases their maximum swimming speed. We propose this high-throughput and reproducible system as an alternative to traditional linear chamber systems for exercising zebrafish and similarly sized fishes.

## INTRODUCTION

Zebrafish are used as a vertebrate model organism for diverse traits in biomedicine, developmental biology, behavioural science and evolutionary biology (*Grunwald & Eisen, 2002*; *Lieschke & Currie, 2007*; *Norton & Bally-Cuif, 2010*). Central to many of these studies is the ability to easily and reliably exercise zebrafish and assay swimming performance (*Palstra et al., 2010*; *Blazina, Vianna & Lara, 2013*; *Gilbert, Zerulla & Tierney, 2014*), especially given that zebrafish models are being increasingly used for understanding the effects of exercise and obesity on human health (*Seebacher et al., 2017*). Furthermore, exercise systems are essential for studies on metabolic performance, where an organism's maximal metabolic rate (MMR) is measured after a period of exhaustive exercise (*Killen, Norin & Halsey, 2016*). It is therefore desirable to have high-throughput and reproducible systems to assess zebrafish swim performance and administer defined exercise regimens.

Common methods to exercise zebrafish include manually chasing fish until exhaustion (*Ferguson & Tufts, 1992*) or traditional Brett-style swimming chambers in which individual fish swim against increasing near-laminar water currents (*Brett, 1964*). These methods

Corresponding author
Shinichi Nakagawa,
s.nakagawa@unsw.edu.au,
itchyshin@gmail.com

are, however, limited by technical requirements of the protocol, low-throughput and/or high cost. For example, studies using exercise respirometry in Brett-style swim chambers typically measure one or two fish per day (*Killen, Norin & Halsey, 2016*). More recent methods, in which zebrafish are forced to swim until exhaustion in a beaker against increasing currents created by a rotating magnetic stir bar (*Blazina, Vianna & Lara, 2013*) generate a large central water vortex, limiting repeatability between experiments.

Here, we present a high-throughput and cost-effective exercise system that uses a coffee plunger to dramatically reduce vortex formation, and illustrate its application to maximal swim performance measurements and exercise training. By reducing, without eliminating, vortex formation we retain the ability to easily score exhaustion during maximal swim performance trials. Our main aim is to demonstrate the use of this new high-throughput system by conducting a pilot study. In this pilot experiment, we assessed (i) intra-class correlation, ICC (in this case, 'intra-individual' correlation or repeatability; *Nakagawa & Schielzeth, 2010*), in estimates of maximum swimming speed, and (ii) whether maximum swimming speed increases after exercise training (e.g., aerobic training paradigm). We additionally investigate whether (iii) maximum swimming speeds differed with sex and length of zebrafish, and (iv) if the mass of zebrafish changed after exercise training.

## MATERIALS & METHODS

### Zebrafish and maintenance

We used 40 (male = 20; female = 20) wildtype zebrafish (*Danio rerio*) that were approximately 6 months old (see below for the rationale for our sample size). Sex was determined from morphology and confirmed with individual egg laying records. Zebrafish were housed in two 3.5 L plastic aquaria supplied with system water (pH = 7.0 to 8.0; water temperature = 27.0 to 29.0 °C) via a filtration system, and placed in a room with a 14.5 h light: 9.5 h dark cycle. Fish were fed once daily in the evening with commercial pellet food (O.range INVE Aquaculture). All experiments were performed with the approval of the Garvan Institute of Medical Research Animal Ethics Committee (Approval number 15/15).

### French press system setup

We set up French press exercise units as illustrated in Fig. 1. Zebrafish swam in the top compartment of the coffee plunger (IKEA Upphetta Coffee/tea maker; 400 mL volume; 8 cm diameter) against circular water currents created by a rotating magnetic stir bar (45 × 7 mm; Fig. 1A). We placed coffee plungers on top of single-plate (Labtek Magnetic Stirrer Hotplate; Fig. 1B) and multi-plate (Labtek 10 Place Stirrer; Fig. 1C) magnetic stirrers for maximal swim performance trials and exercise training, respectively. Since swim performance is shown to be sensitive to water temperature (*Wardle, 1980*), we recorded water temperature for each trial and replaced water as necessary between trials to keep water temperature constant (average water temperature across trials = 25.5 ± 0.1 °C). Full protocols and a video illustration of the system are available in the Supplementary Information.

Preliminary trials indicated that the water speed in the zebrafish swimming compartment was substantially slower than the set speed of the magnetic stirrer. We thus determined

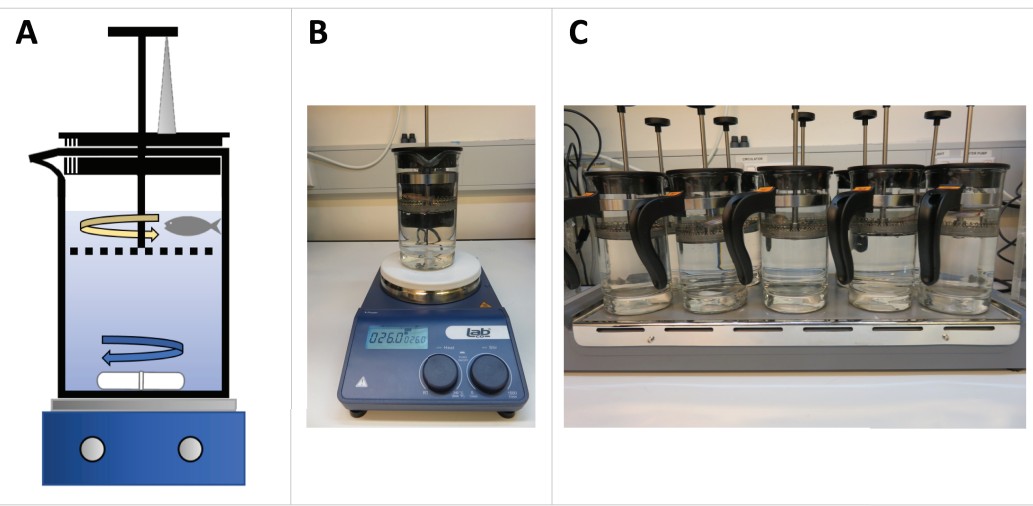

**Figure 1** **French press exercise system setup for maximal swim performance trials and exercise training of zebrafish.** (A) Diagram of French press unit setup. Coffee plungers were 75% filled with water and placed on a magnetic stirrer. Zebrafish swam against water currents (yellow arrow) generated by a rotating magnetic stir bar (blue arrow). Plunger height was standardized within and between trials by propping up the plunger using a vertical pipette tip (grey triangle). (B) A single-plate coffee plunger set up for quantifying maximum swimming speed. The stirrer speed was increased incrementally until maximum swimming speed was reached (range: 360–910 rpm) (C) Multi-plate coffee plunger setup for multiplexed exercise training. Up to ten zebrafish were simultaneously exercised in separate coffee plungers at a submaximal speed.

actual water speeds experienced by the zebrafish by tethering a freely rotating plastic leaf above the plunger and determined its rpm for a range of stirrer speeds (200–1,400 rpm). Stirrer speed and water speed exhibited a linear relationship (Fig. S3).

## Maximal swim performance

We measured individual maximum swimming speeds of 40 adult zebrafish. We gradually increased the stirrer speed at 10 rpm steps every 5 s until maximum swimming speed, defined as the speed at which zebrafish lost their ability to maintain their position in the water column. Importantly, our step duration was long enough to observe the recovery of zebrafish that were temporarily swept up into the water vortex at sub-maximal speeds, and thus avoid underestimation of maximum swimming speeds. To assess individual repeatability and the effect of exercise training, maximum swimming speed was measured twice for each zebrafish (i.e., pre- and post-exercise training; an outline of the experimental design is shown in Fig. S4). One day after the first round of maximal swim performance trials, each zebrafish ($n = 40$) was randomly assigned to a control ($n = 20$) or experimental group ($n = 20$) that were matched for pre-intervention maximum swimming speeds. Both our first and second round of maximal swim performance trials were conducted blind to the non-exercised (control) and exercised (experimental) groups. All maximal swim performance trials were conducted between 9 am and 3 pm.

Body mass and length of each zebrafish were also measured before each of the two maximal swim performance trials. We weighed individual zebrafish by placing each
fish into a small, transparent holding container filled with system water and placed atop a measuring scale (OHAUS Adventure Pro Precision Electronic Balance). To measure length, the holding container was then placed on top of graphing paper (1 mm$^2$ grid squares) and photographs of each fish were taken directly from above with a digital camera (Canon PowerShot SX710HS). The body length of each zebrafish was estimated by calibrating individual photographs using ImageJ (version 1.51 k; *Schneider, Rasband & Eliceiri, 2012*).

### Exercise training

To investigate the effects of exercise training, the experimental group received a 40-minute swimming regime at a sub-maximal exercise speed (stirrer speed = ~370 rpm) for five consecutive days using multi-plate magnetic stirrers. The control group was placed in identical French press units for the same duration without any forced exercise (stirrer speed = 0 rpm). Exercise training aimed to improve swimming ability (*Brett, 1964*; *LeMoine et al., 2010*) and the sub-maximal swimming speed was determined during preliminary trials as the highest tolerated continuous swimming speed that did not induce exhaustion within 40 min. All exercise training trials were conducted at 9 am.

### Statistical analysis

Our sample size of 40 fish (male = 20; female = 20) was determined by balancing logistical considerations (e.g., the use of 10-plate magnetic stirrer, meaning 10 fish can be exercised at one time) and with reasonable power. Our design only allowed us to detect medium to large effects (Cohen's $d = 0.45$; note $d = 0.3$, $0.5$ and $0.8$ represents small, medium and large effects; *sensu Cohen, 1988*) setting the power at 0.8 and with the alpha level of 0.05. All power calculations were conducted using the *pwr* package (*Champely, 2017*).

   We estimated repeatability (ICC estimates) in maximum swimming speeds using the *rpt* function in the package *rptR*, which estimates repeatability using a linear mixed-effects model (LMM) framework (*Stoffel, Nakagawa & Schielzeth, 2017*). Repeatability ($R$) describes the proportion of among-group ($\sigma_\alpha^2$) variance out of the total variance (*Sokal & Rohlf, 1995*):

$$R = \sigma_\alpha^2/(\sigma_\alpha^2 + \sigma_\varepsilon^2).$$

We obtained three different estimates of repeatability, namely unadjusted, adjusted, and enhanced agreement repeatabilities, using separate models for each. Unadjusted repeatability (also referred to as agreement or broad sense repeatability; *Shrout & Fleiss, 1979*) does not control for variance explained by fixed effects in the repeatability estimate, and in our case, only included individual fish ID as a random effect. Moreover, adjusted repeatability accounts for fixed effects by excluding the variance explained by fixed effects from the repeatability estimate, whilst enhanced agreement repeatability accounts for fixed effects by including their variance in the estimate (*Nakagawa & Schielzeth, 2010*; *Stoffel, Nakagawa & Schielzeth, 2017*). We included sex and length of zebrafish, as well as an interaction between treatment and measure as our fixed effects (see Table S1 for coding of our variables). Uncertainty in repeatability estimates was obtained using parametric bootstrapping, which was set to 10,000 for all models.

**Table 1** **Model coefficients from linear mixed-effects model investigating differences in maximum swimming speed (rpm).** Fixed effect intercept represents (i) first measure, (ii) experimental group, and (iii) female sex, mean-centered for zebrafish length and water temperature. Slope estimates, 95% lower (LCI) and upper (UCI) confidence intervals, $t$-values ($t$), degrees of freedom ($df$) and P-values ($P$) are reported.

| Fixed effects | Estimate | LCI | UCI | $t$ | $df$ | $P$ |
|---|---|---|---|---|---|---|
| Intercept | 614.3 | 558.9 | 669.7 | 20.982 | 55.90 | <0.001 |
| Measure (second) | 77.5 | 21.6 | 134.4 | 2.665 | 37.87 | 0.011 |
| Treatment (control) | 20.7 | −50.5 | 91.8 | 0.549 | 61.42 | 0.585 |
| Sex (male) | 34.3 | −26.5 | 95.2 | 1.070 | 35.37 | 0.292 |
| Length | 8.6 | −3.7 | 20.9 | 1.326 | 35.38 | 0.193 |
| Measure by treatment (second*control) | −41.0 | −119.1 | 38.3 | −1.009 | 37.41 | 0.319 |
| Water temperature | −2.9 | −44.4 | 36.4 | −0.140 | 58.58 | 0.889 |

| Random effects | Variance ($\sigma^2$) | SD ($\sigma$) | LCI ($\sigma$) | UCI ($\sigma$) |
|---|---|---|---|---|
| Individual fish ID | 5,416 | 73.6 | 32.2 | 100.0 |
| Residual | 7,969 | 89.3 | 70.1 | 109.4 |

To analyse the effect of exercise training on maximum swimming speed, we constructed linear mixed-effects models using the *lmer* function in the *lme4* package (*Bates et al., 2015*) fitting individual fish IDs as a random effect. For fixed effects, we fitted an interaction between treatment and measure to assess whether exercise training increased swim speed in the second round of the maximum swim performance trial. We further included as fixed effects sex and length of individual fish (to investigate differences in maximum swim speed with zebrafish sex and length) and water temperature (to control for any differences due to water temperature). Length and temperature variables were mean-centered prior to analysis. To assess change in mass after exercise training, in a separate model we fitted sex of zebrafish (to account for sex differences in mass), and an interaction between treatment and measure as fixed effects. We assessed statistical significance (at the alpha level = 0.05) using Satterthwaite's approximation for degrees of freedom obtained from the package *lmerTest*. Confidence intervals for variables were obtained using the *confint* function provided in the *lme4* package. All statistical analyses were conducted in the R programming language (v 3.2.2; *R Development Core Team, 2016*; see Supplementary Information for data and analyses).

## RESULTS

Maximum swimming speed significantly increased in the exercise-trained group, but was unchanged in the control group (Exercise: $t = 2.67$, $df = 37.87$, $P = 0.011$; Control: $t = 1.15$, $df = 40.71$, $P = 0.256$; Table 1; Fig. 2). Repeatability estimates offer important insights into the consistency of phenotypes and measurement accuracy (*Nakagawa & Schielzeth, 2010*; *Stoffel, Nakagawa & Schielzeth, 2017*). We observed moderate ICC estimates of maximum swimming speeds after accounting for differences between control and treatment groups (point estimate range: 0.34–0.41; Fig. 3).

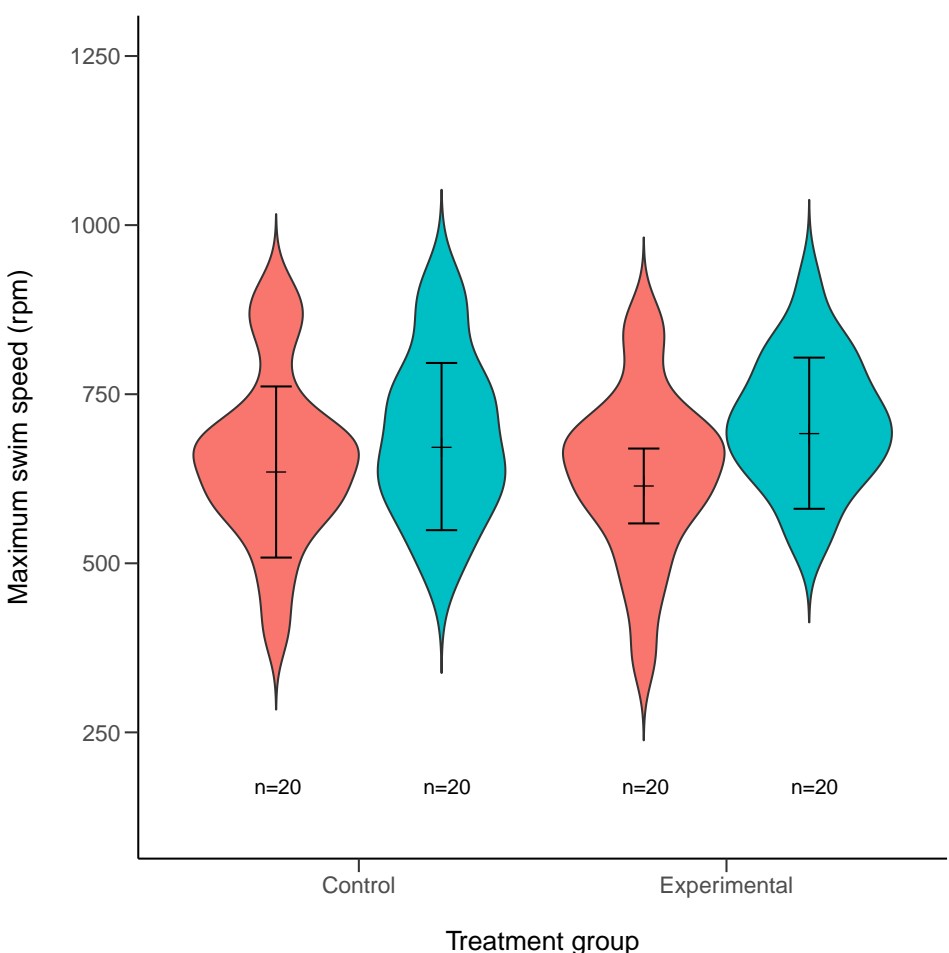

**Figure 2** **Violin plots of maximum swimming speed for experimental (exercised) and control (non-exercised) groups.** Violin plots show the distribution of individuals maximum speed estimates obtained from the first (red) and second (green) round of maximum swim performance trials for both experimental and control groups (sample sizes (n) are given for each group below violin plots). Arrows inside the violin plots represent mean estimate of maximum swim speeds and their 95% confidence intervals, as obtained from linear mixed-effects models. Maximum swim speed increased significantly in the second round (green) of maximal swim performance trials compared to the initial round (red) in the exercise trained (experimental) group ($t = 2.67$, $df = 37.87$, $P = 0.011$). Swim speed also increased slightly in the non-exercised control group, although this increase was not statistically significant ($t = 1.15$, $df = 40.71$, $P = 0.256$).

We found weak evidence that maximum swimming speed was impacted by the sex or the length of zebrafish (Table 1). We also did not find differences in mass for the exercise-trained group compared to the control, although mass decreased slightly in the second measure of exhaustive exercise relative to the first ($t = -2.15$; $df = 38.00$; $P = 0.038$; Table S2).

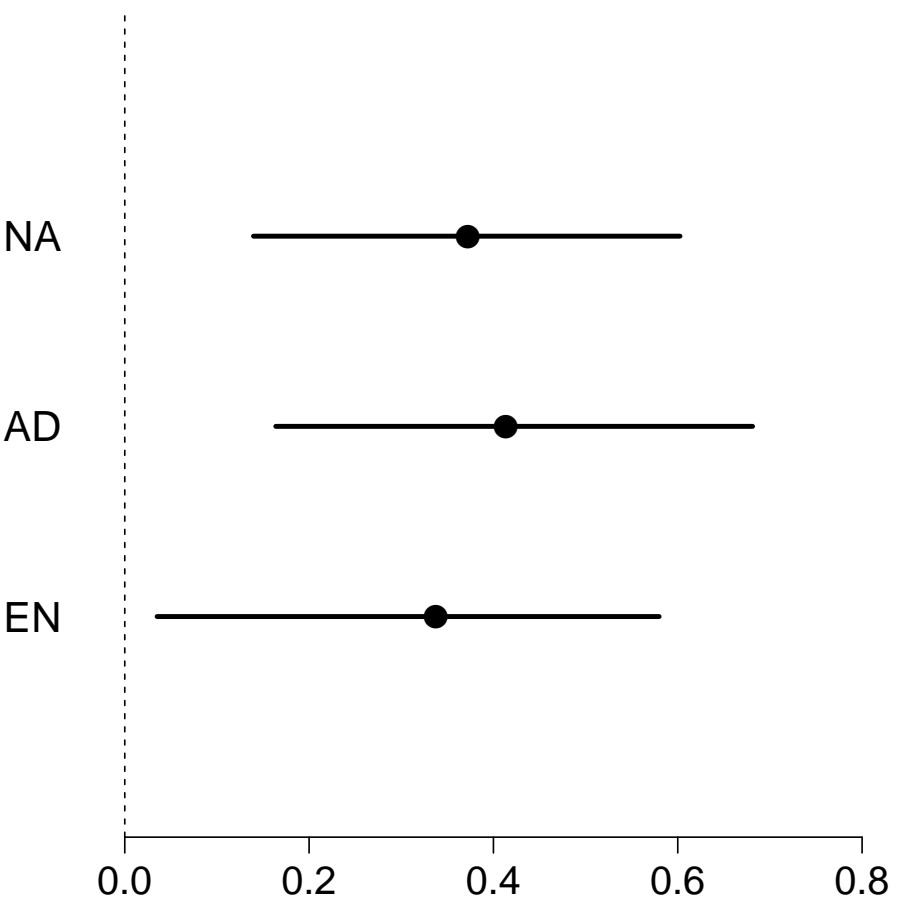

**Figure 3 Forest plot of repeatability estimates in maximum swimming speed.** Repeatability estimates (circles) are plotted with 95% confidence intervals (lines). Estimates of unadjusted (NA; 0.34, 0.04 to 0.58), adjusted (AD; 0.41, 0.16 to 0.68) and enhanced agreement (EN; 0.37, 0.14 to 0.60) repeatabilities (or intra-class correlations, ICC) show that maximum swimming speed is significantly repeatable.

## DISCUSSION

Our main aim in this study was to demonstrate a new method that can be used for exercising and measuring swim performance in zebrafish. Protocols to exercise and assess swim performance in zebrafish have broad applications including studies of human disease (*Lieschke & Currie, 2007*), metabolism (*Killen, Norin & Halsey, 2016*) and behaviour (*Norton & Bally-Cuif, 2010*). In our pilot study, we demonstrated that we could obtain repeatable estimates of maximal swim performance. The use of a multi-plate magnetic stirrer array further enabled the high-throughput administration of an exercise regimen in a standardized manner. Additionally, we found exercise training to increase maximum swimming speed, which may reflect improved swimming capacity due to physiological and/or behavioural plasticity (*Davison, 1997*; *Sinclair et al., 2014*). We acknowledge that our pilot experiment had low power to detect any effects below $d = 0.45$. Therefore, interest in sex and size effects warrant further investigation. Further, our finding of exercise eliciting

an increase in swim performance should be replicated in the future when researchers employ our proposed method.

The simplicity of our maximal swim performance protocol allowed an observer to simultaneously test two zebrafish. With a 10-plate set-up, for example, the maximum swimming speed of ~240 zebrafish can be assessed in 5 h with three observers (~7.5 min for each measurement). The duration of our maximal swim performance test falls between commonly defined $U_{crit}$ and burst swimming performance tests (*Brett, 1964*; *Tierney, 2011*), and thus swimming capacity as captured in our protocol may reflect a mixture of anaerobic and aerobic swimming abilities. By adjusting the magnitude and duration of flow speed it is possible to capture swimming abilities underlying different metabolic, physiological and ecological interests (*Tierney, 2011*). Our French press exercise system provides additional flexibility through adjustable temperature control settings. Understanding temperature related effects on exercise and swimming performance may be particularly useful in the context of organismal responses to climate change (*Clark, Sandblom & Jutfelt, 2013*).

While our high-throughput and cost-effective protocol has clear advantages over current methods, it has three potential limitations. First, water speed varies across the radius of the coffee plunger potentially causing heavier fish to lose control at lower speeds than in a comparable linear swimming chamber (*Rummer et al., 2016*). This process would contribute a mass and position specific error for a given trial, possibly reducing repeatability. Despite this, we obtain moderate and significantly repeatable ICC estimates that are comparable to previous estimates of average repeatability in behavioural traits (*Bell, Hankison & Laskowski, 2009*; *Holtmann, Lagisz & Nakagawa, 2017*), suggesting that this error is unlikely to compromise the utility of the system as a general method to quantify maximal swimming performance. Moreover, similar repeatability estimates obtained from both unadjusted and adjusted repeatability models suggest mass specific error to be negligible. Matching the size of the French press swimming chamber to the particular species of interest may also help reduce position specific error. Second, imbalanced use of one side of the fish musculature due to circular swimming may result in premature, rather than complete, exhaustion (*Nilsson et al., 2007*). Swim training in both clockwise and anti-clockwise directions, as permitted with many standard magnetic stirrers, may be required to test swim performance for both sides of the fish musculature. Finally, when considering endpoints related to swim performance, it is important to consider that the kinematics of circular swimming will be different than that of swimming in linear systems (*Drucker & Lauder, 1999*). Nonetheless, the simplicity of the proposed exercise system and its compatibility with multi-plate magnetic stirrers allows for a high-throughput and repeatable exercise protocol in zebrafish, and offers an affordable alternative to traditional linear systems.

## CONCLUSION

In this study, we have presented a novel and high-throughput fish exercise system that provides several advantages over using traditional linear chamber systems. Through the use of a coffee plunger and magnetic stirrer, we are able to run a variety of exercise regimes

with relative ease and obtain repeatable and comparable estimates of swim performance. We propose that this simple and reproducible method will be useful in a variety of research fields increasingly using zebrafish and other species of fish for exercise, including in medical, metabolic and behavioural research.

## ACKNOWLEDGEMENTS

We thank the staff at Garvan Institute of Medical Research for their support and general husbandry of zebrafish. We additionally thank Harry G. Thomas for help during trials. We acknowledge that comments from the two reviewers and the editor significantly improved earlier versions of this manuscript.

### Funding

Shinichi Nakagawa is supported by Future Fellowship (FT130100268) and a start-up fund from UNSW. Daniel W.A. Noble was supported by an ARC Discovery Early Career Research Award (DE150101774) and a UNSW VC Research Fellowship. Daniel Hesselson was supported by NHMRC Project Grants (GNT1063981 and GNT1130222). The funders had no role in study design, data collection and analysis, decision to publish, or preparation of the manuscript.

### Grant Disclosures

The following grant information was disclosed by the authors:
Future Fellowship: FT130100268.
UNSW.
RC Discovery Early Career Research Award: DE150101774.
UNSW VC Research Fellowship.
NHMRC Project Grants: GNT1063981, GNT1130222.

### Competing Interests

The authors declare there are no competing interests.

### Author Contributions

- Takuji Usui conceived and designed the experiments, performed the experiments, analyzed the data, contributed reagents/materials/analysis tools, wrote the paper, prepared figures and/or tables, reviewed drafts of the paper.
- Daniel W.A. Noble and Malgorzata Lagisz performed the experiments, contributed reagents/materials/analysis tools, prepared figures and/or tables, reviewed drafts of the paper.
- Rose E. O'Dea and Melissa L. Fangmeier performed the experiments, contributed reagents/materials/analysis tools, reviewed drafts of the paper.
- Daniel Hesselson and Shinichi Nakagawa conceived and designed the experiments, performed the experiments, contributed reagents/materials/analysis tools, reviewed drafts of the paper.

## Animal Ethics

The following information was supplied relating to ethical approvals (i.e., approving body and any reference numbers):

This work was conducted with permission from the Garvan Institute Animal Ethics Committee (Approval number 15/15).

## Data Availability

The raw data and R code have been provided as Supplemental Files.

## Supplemental Information

Supplemental information for this article can be found online at http://dx.doi.org/10.7717/peerj.4292#supplemental-information.

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
