# Peer review of "The French press: a repeatable and high-throughput approach to exercising zebrafish (Danio rerio)"

_PeerJ, doi:10.7717/peerj.4292_

## Round 0.1 · original submission · Major Revisions

Your manuscript entitled "The French press: A repeatable and high-throughput approach to exercising zebrafish" has now been reviewed and the reviewers comments are appended below. You will see that, while they find your work of interest, they have raised several points that need to be addressed by a major revision.

I therefore invite you to revise and resubmit your manuscript, taking into account the points raised.

·

Basic reporting

The study proposes in a clear and a well-grounded way a novel, cost-effective and high-throughput method for exercising zebrafish, using a coffee plunger and magnetic stirrer. The introduction has a good scientific background and makes the study’s aim clear, in addition to being well-contextualized regarding the literature. The study was developed and discussed so that it answers the main questions that were proposed. The structure of the text and the format of the figures are in accordance with this journal’s norms and the authors supplied enough data to understand the results analysis. There are, however, some minor shortcomings as follows:

Experimental design

a. Bearing in mind that this study aims to standardize a new technique using animals and this kind of study is very sensitive to various interfering factors, it is paramount to add a “Zebrafish maintenance” paragraph in the “Materials and Methods (line 70)” section, detailing: the strain, age and nourishment of the zebrafish; the water control parameters; light/dark cycle. It is also important to add the ethics committee approval for the use of these animals in the experiments.
b. Please include the time of the day in which the “Maximal swim performance (line 90)” and “Exercise training (line 101)” were conducted.
c. Please add the French Press brand used in the tests (line 72).
d. In the “Maximum swim performance” section (line 90), what was the method of weighing, sexing and measurement of the animals?
e. In the “Exercise training” section (line 101), how long after the “maximal swim performance” test were the animals taken to the following “pre-intervention maximum swimming speeds” test (hours, days etc.)?
f. The authors should provide a better explanation on how the effects of exercise training were assessed (line 149-151), at the end of the 5 days of the swimming regime at a sub-maximal exercise speed, to evaluate the difference in relation to the control group.
g. As the study has different stages, the methodology would be clearer if a figure were added with a timeline to better express the events, at the authors’ discretion.

Validity of the findings

a. Both the statistical analysis and graphs seem adequate for this type of study.
b. In the results section is described “Maximum swimming speed significantly increased in the exercise-trained, but was unchanged in the control group (lines 149-150)”. It would be interesting to include a graph of these data.
c. Although the authors wrote a conclusion at the end of the discussion (lines 201-203), it would be appropriate to emphasize the conclusion, by writing a conclusion section in a new paragraph.

Additional comments

Taking into consideration that this study proposes an innovative method that could be used in different approaches, I consider this article relevant to be published after improvements are made.

Reviewer 2 ·

Basic reporting

The manuscript describes improvements to existing exercise protocols for zebrafish and proceeds to investigate the repeatability of the measurement approach. The manuscript also reports to investigate improvements in maximum speed following exercise training, mass increases following training, and the effects of sex and length on the results. Most of the results are missing from the manuscript and those that are present are largely reported in supplementary tables.

1) In order to address this the results relevant to each of the questions listed above need to be included in the main text.

2) Data investigating the similarities between different plungers can be moved to the supplementary data

Experimental design

In the methods it is reported that 40 fish were measured.

1) Given that multiple questions were asked this should be further explained to give the n number for each experiment. For Example to look at the effects of sex and exercise there was only 10 fish in each group.

2) A single experiment with n=10 was carried out to address the questions raised. It is not clear if this is sufficient power to detect the effects proposed. The authors need to justify the size of the experimental group based on power analysis to indicate the level of change in swimming speed, mass etc, they are able to detect If the number of fish is insufficient the experiments should be repeated with greater numbers to support the findings.

3) The repeatability was assessed by using measurements pre and post-exercise training. It is not possible to use the same data to assess repeatability as the training could induce a difference. Repeatability should only utilise the unexercised fish.

4) For the analysis of water speed and stirrer speed using different French press units, 2 units is not sufficient to say there is no variation between units, and additional measurements are required. Standard practice would suggest a minimum of 3 but given that the authors have access to many more they are encouraged to increase the number significantly.

5) The make and model of the coffee plunger used should be supplied in the methods section.

Validity of the findings

The authors introduce a useful advance on the previous exercise approaches for zebrafish. The conclusions regarding the effects of exercise however, are questionable given the low number of fish utilised and the lack of experimental repetition.

It will be vital to demonstrate that the experiments were sufficiently powered to address the questions raised to demonstrate the validity of the findings.

---

## Round 0.2 · Major Revisions

I regret to inform you that based on the reports and comments received from the referees, I am unable to accept the above-mentioned manuscript for publication in PeerJ in this present form. Although one reviewer has been favorable, other clearly highlighted important points in the article that supported my decision. In this way, I suggest to the authors include in the manuscript some paragraph(s) of limitations based on referee comment.

·

Basic reporting

No comment.

Experimental design

No comment.

Validity of the findings

No comment.

Additional comments

Thank you for making the proposed changes.
I consider that your article is now much more clear and elucidative.
Therefore, the article is ready for publication.

Reviewer 2 ·

Basic reporting

The new figure (Fig2) presenting the change in swim speed with exercise is a very useful addition. However, measurements from individual fish should be presented to allow visualisation of the variation, which is currently hidden by only presenting the estimate from the mixed model? (presumed, not given in figure legend) and standard error bars rather than standard deviation or confidence intervals.

Experimental design

The reviewers have made some changes as a response to the previous review but some key concerns remain to be addressed.

The clarification of n numbers and variables in the mixed model is an improvement.
However, the response to the question of sample size has not addressed the concern.

The authors are correct to suggest that post-hoc power analysis are not ideal, but it is still critical to ensure that sufficient animals have been examined. The authors themselves have previously published on the important of prospective power analysis, and if they are not willing to conduct a post-hoc analyses then details of the prospective power that determined that n=20 would be sufficient.

The authors also state that the sample size was sufficient to detect an improvement in swimming speed in the experimental group. This does not indicate that sufficient numbers were present in the case of sex, length, or mass where the effect may be less. Furthermore, for the effect of sex on exercise response there were only 10 exercised males and 10 exercised females (n=10).

Identifying issues with post-hoc analyses is not sufficient address the concern that the number of animals in this study is probably too low to identify effects which must be addressed. The lack of experimental repetition has also not been addressed and there is no indication in the methods that experimenters were blinded to treatment group.

Validity of the findings

Given the lack of justification of the number of animals examined, lack of experimental repetition, and the fact that observers were not blinded to treatment group there are still concerns over the validity of findings.

Additional comments

Overall this is an interesting method that is of interest to the field but the examination of the effect of exercise on maximal swimming speed is not conducted with sufficient numbers and experimental repetition to address the questions raised.

---

## Round 0.3 · accepted · Accept

I am pleased to inform you that your manuscript referenced above has been accepted for publication in PeerJ.